# Dissolved Carbon Concentrations and Emission Fluxes in Rivers and Lakes of Central Asia (Sayan–Altai Mountain Region, Tyva)

Arisiya A. Byzaakay [1,2], Larisa G. Kolesnichenko [1], Iury Ia. Kolesnichenko [1], Aldynay O. Khovalyg [2], Tatyana V. Raudina [1], Anatoly S. Prokushkin [3], Inna V. Lushchaeva [1], Zoia N. Kvasnikova [4], Sergey N. Vorobyev [1], Oleg S. Pokrovsky [5,6,*] and Sergey Kirpotin [1]

1 Bio-Clim-Land Centre of Excellence, Tomsk State University, Lenin Avenue, 36, 634050 Tomsk, Russia; arisiy@inbox.ru (A.A.B.); klg77777@mail.ru (L.G.K.); vancansywork@mail.ru (I.I.K.); tanya_raud@mail.ru (T.V.R.); luschaeva@mail.ru (I.V.L.); soil@green.tsu.ru (S.N.V.); kirp@mail.tsu.ru (S.K.)
2 Research Organization Department, Tuvan State University, Lenina Street, 36, 667000 Kyzyl, Russia; aldyn@mail.ru
3 V.N. Sukachev Institute of Forest SB RAS, Akademgorodok 50/28, 660036 Krasnoyarsk, Russia; prokushkin@ksc.krasn.ru
4 Geology and Geography Faculty, Tomsk State University, Lenin Avenue, 36, 634050 Tomsk, Russia; zojkwas@rambler.ru
5 Geosciences and Environment Toulouse, UMR 5563 CNRS, 31400 Toulouse, France
6 N. Laverov Federal Center for Integrated Arctic Research of the Ural Branch of the Russian Academy of Sciences, Northern Dvina Emb. 23, 163000 Arkhangelsk, Russia
* Correspondence: oleg.pokrovsky@get.omp.eu; Tel.: +33-561332625

**Abstract:** The carbon (C) cycle in inland waters, including carbon concentrations in and carbon dioxide ($CO_2$) emissions from water surfaces, are at the forefront of biogeochemical studies, especially in regions strongly impacted by ongoing climate change. Towards a better understanding of C storage, transport and emission in Central Asian mountain regions, an area of knowledge that has been extremely poorly studied until now, here, we carried out systematic measurements of dissolved C and $CO_2$ emissions in rivers and lakes located along a macrotransect of various natural landscapes in the Sayan–Altai mountain region, from the high mountains of the Western Sayan in the northwest of Tyva to the arid (dry) steppes and semideserts in the intermountain basins in the southeast of Tyva on the border with Mongolia. New data on major hydrochemical parameters and $CO_2$ fluxes ($fCO_2$) gathered by floating chambers and dissolved organic and inorganic carbon (DOC and DIC, respectively) concentrations collected over the four main hydrological seasons allowed us to assess the current C biogeochemical status of these water bodies in order to judge possible future changes under climate warming. We further tested the impact of permafrost, river watershed size, lake area and climate parameters as well as 'internal' biogeochemical drivers (pH, mineralization, organic matter quality and bacterial population) on $CO_2$ concentration and emissions in lakes and rivers of this region and compared them with available data from other subarctic and mountain settings. We found strong environmental control of the $CO_2$ pattern in the studied water bodies, with thermokarst lakes being drastically different from other lakes. In freshwater lakes, $pCO_2$ negatively correlated with $O_2$, whereas the water temperature exerted a positive impact on $pCO_2$ in large rivers. Overall, the large complexity of counteracting external and internal drivers of $CO_2$ exchange between the water surfaces and the atmosphere ($CO_2$-rich underground DIC influx and lateral soil and subsurface water; $CO_2$ production in the water column due to dissolved and particulate OC biodegradation; $CO_2$ uptake by aquatic biota) precluded establishing simple causalities between a single environmental parameter and the $fCO_2$ of rivers and lakes. The season-averaged $CO_2$ emission flux from the rivers of Tyva measured in this study was comparable, with some uncertainty, to the C uptake fluxes from terrestrial ecosystems of the region, which were assessed in other works.

**Keywords:** carbon; concentration; $CO_2$; emission; rivers; lakes; biogeochemical cycle; Central Asia; Altai–Sayan Mountains

## 1. Introduction

The carbon (C) cycle in inland waters, including the dissolved and particulate concentrations of carbon and the carbon dioxide ($CO_2$) emissions, are at the forefront of biogeochemical studies, especially in the regions most sensitive to ongoing climate change such as boreal and subarctic zones [1–10] or the tropical/equatorial belt [11–14]. In contrast to these numerous works, C storage, transport and emission in central continental, mountain and arid regions remain strongly understudied, due either to limited access and logistics or to the still underestimated potential role of these remote territories in C cycling in inland waters. This is especially true for the Central Asian mountain system encompassing the Tibet, Himalaya, Pamir, Altai and Sayan regions. Exceptions are the thorough works on DOC, DIC and POC fluxes in the Himalayan rivers [15–17] and C concentrations and fluxes in thermokarst lakes of the Tibetan Plateau [18–23]. However, the northern part of the Central Asian Mountain System remains virtually unexplored from the viewpoint of hydrochemistry and carbon balance in its rivers and lakes.

The Altai–Sayan Mountain system is a specific inland region of the northern part of Central Asia which covers the territory of four countries: Russia, Mongolia, China and Kazakhstan [24]. In Russia, it is located within the Tyva Republic, Altai Republic, Buryatia, Khakassia, and Krasnoyarsk Krai. It is characterized by a high degree of continentality and aridity as well as the highest level of endemicity under a huge variety of ecosystems and landscapes, many of which are vulnerable to climatic changes [25]. The water bodies such as rivers and lakes are of particular importance for the sustainability, ecosystem services and conservation of the biodiversity of this arid region [26–31]. The particular interest in the Altai–Sayan region is that currently occurring climate changes in this territory contradict existing world models and forecasts. In particular, in the Tyva Republic, our group reported the first ever observed occurrence of the natural phenomenon of 'greening' or afforestation of steppes and bare sands [32,33]. This finding contrasts with a number of prediction models that stipulated further progressive drying of the arid Altai and Sayan regions (ASR), such as Khakassia, Tyva and Mongolia [34,35]. The other existing assessments of future climates in arid regions of Eurasia also predict an increase in aridity and even propose the propagation of desertification in the steppe regions [36–38]. For instance, in the Tyva Republic, the areal extension of steppe ecosystems, including dry steppes, is predicted to increase by 20–65%, whereas in semideserts, the projected increase is predicted to be several hundred percent more compared to their current extent [24].

In contrast to the above-mentioned predictions, some other studies suggest an increase in the climatic instability, such as drastic transformation of the main atmospheric circulation, in the south Siberia regions and at the border with Central Asia, which could lead to a decrease in the role of atmospheric transport from Atlantic regions, resulting in the blocking of anticyclones and an increase in meridional atmospheric transport [39,40]. This process, in turn, could cause many catastrophic weather events, such as the dramatic appearance of rainstorms, hurricanes and water floods, together leading to the progressive humidification of arid, previously dry regions [41]. Such catastrophic weather events have been observed over the last few years in Khakassia, Tuva and Mongolia [42–44].

It is clear that the ongoing climate changes can drastically impact the current hydrochemical, biogeochemical and hydrological status of the inland rivers and lakes of the region. For example, it is known that lakes at high altitudes and cold climates are particularly sensitive to global change [45,46] and modifications in carbon biogeochemistry, including the status of dissolved organic matter (DOM), and C emission fluxes may strongly alter the role of these lakes in the global C cycle [47–49]. Among the consequences of the progressive humidification of Central Asian arid regions under ongoing climate change, the treeline shift (greening of upland and advancement of treeline) can strongly affect the biogeochemical functioning of lakes, given that 15% of all lakes globally are located at elevations 500 m or more above sea level [50]. Further, existing paleo-reconstructions suggest that lake productivity [51] and biogeochemistry [52] are sensitive to changes in DOM input linked to treeline position. Therefore, thorough assessment of today's status

of the C biogeochemical cycle (concentration, emission from the water surfaces) is necessary to be able to judge the possible future changes in Central Asian regions induced by climate instability.

Over the past decade, a few studies in this area addressed the hydrochemical [53–58], hydrological and hydrographic [59–61] status of the water bodies, including the balneological aspect of mineral springs [62–65]. There are also some studies conducted on the hydrochemistry of water bodies in adjacent regions of Mongolia [66]. However, the status of aquatic C and $CO_2$ emissions from the water surfaces of the Altai–Sayan region remain unknown. Towards filling this gap in knowledge, here, we assessed the concentration of dissolved organic and inorganic carbon and C emissions in lakes and rivers of the Tyva region by selecting large and small water bodies affected by permafrost to different degrees. A novel physicogeographical and climatic transect of inland water bodies, which we implemented in this study, extends from the northwest (the highlands of the western Sayan) to the southeast (the semideserts of the Ubsunur Depression on the border with Mongolia). This transect comprises a large variety of natural ecosystems and landscapes of the region, from glacial–nival high mountain belts to foothill taiga forests, intermountain basins with mixed herb and steppe ecosystems and dry semideserts. As a working hypothesis, we anticipated strong environmental control of C biogeochemical parameters of lakes and rivers, including climate, altitude, permafrost extent and size of the watershed as the main 'external' drivers of C concentration and emission. We also assessed the link between carbon parameters of the water bodies and possible 'internal' biogeochemical drivers (hydrochemical parameters) such as pH, mineralization, quality of dissolved organic matter and bacterial abundance. We tested these controls across the four main hydrological seasons (spring, summer, autumn and winter) in five lakes and 15 rivers of different sizes and landscape contexts.

## 2. Materials and Methods

### 2.1. Water Bodies of the Tyva Republic

We visited 20 water bodies during four hydrological seasons (autumn 2021–summer 2022), as shown in Figures 1 and 2. The main physicogeographical parameters of the rivers and lakes are listed in Tables 1 and 2 and are described in detail in the Appendix A. The Republic of Tyva is located in the Sayan–Altai Mountain region, between 50 and 53° N and between 88 and 99° E; the elevations range from 2000 to 500 m, creating a large variety of landscapes, from high-altitude belts to basins with steppes and semideserts. The climate of the region is continental, with cold, long winters and hot summers; the mean monthly temperatures range from −41 in January to +35 °C in July; the precipitation is low (115 to 350 mm y$^{-1}$), and about 70% of it falls during the warm season of the year [67].

During the period of our study, the temperature ranged from −28 °C to 21 °C, and the precipitation was equal to 322 and 144 m during 2021 and 2022, respectively. (http://www.pogodaiklimat.ru/history/36096_2.htm, accessed on 23 September 2023). The water objects sampled in this work include numerous water bodies mainly belonging to the Yenisei River basin, whereas only a small number belonged to the closed basin of the Ubsunur basin.

The largest river flowing through the region is the Yenisei River, formed at the confluence of the Maly and Bolshoy Yenisei rivers. Most rivers of the Yenisei basin are of mountain origin, that is, they mainly have snow and groundwater feeding. The studied lakes are mostly drainless and mainly fed by groundwater, with the exception of Lake Chagytai [68,69]. We collected the water samples during four hydrological seasons, in autumn (24 October 2021–27 October 2021), winter (7 March 2022–11 March 2022), spring (18 May 2022–22 May 2022) and summer (19 August 2022–23 August 2022). During each season, we sampled 20 water bodies (15 rivers and five lakes), with 140 samples in total. These were selected according to the criterion of proximity to the weather stations throughout the climate/landscape macrotransect of Tyva.

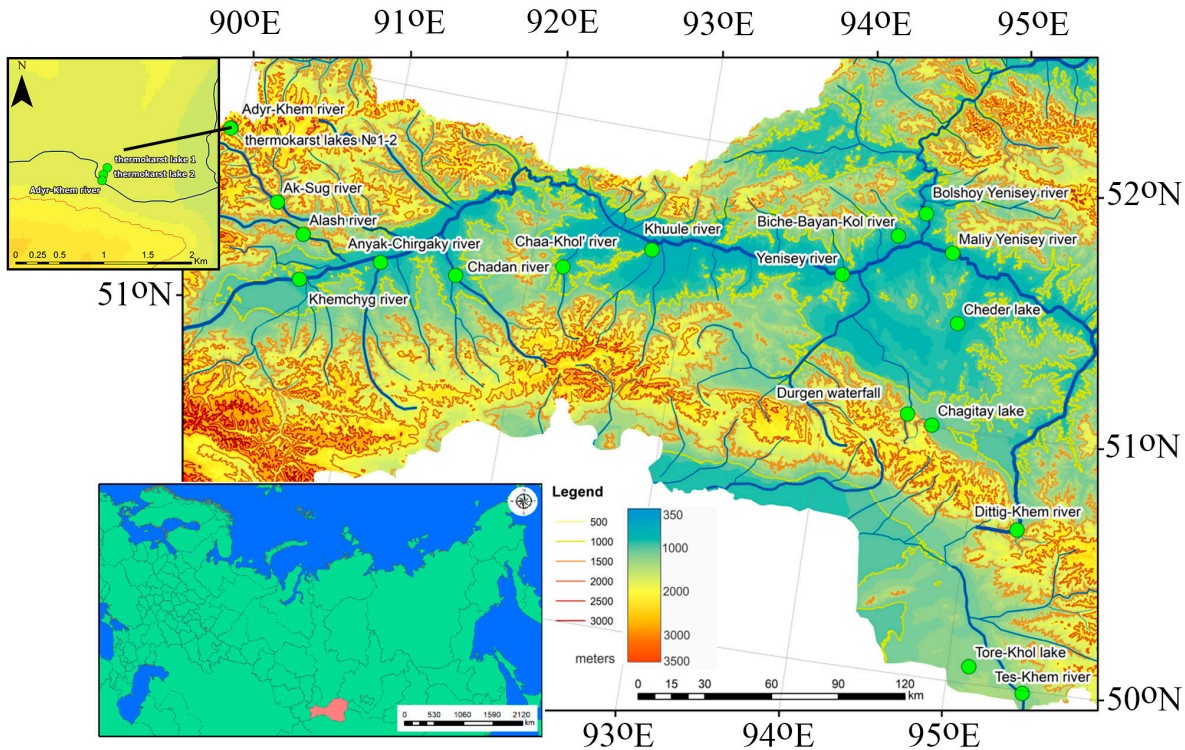

**Figure 1.** Location of research objects within the Sayan–Altai Mountain system (Central Asia).

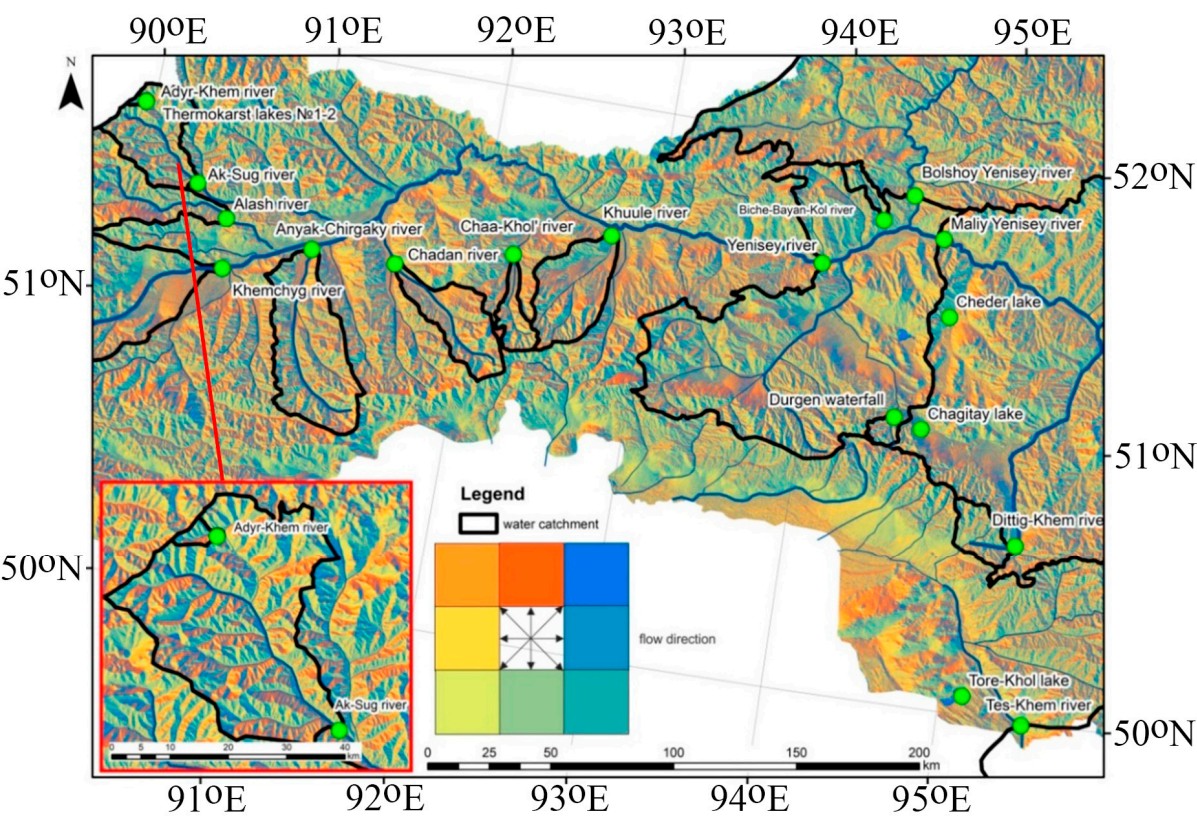

**Figure 2.** River catchments of the study region. The direction of water flow is shown by different colors.

**Table 1.** General information about the rivers sampled in this work.

| Name | Flow Velocity m/s | Depth (m) | Length (km) | Average Discharge, $m^3/c$ | Catchment Area $(km^2)$ | Height at the Samp-Ling Point, m | Average Catchment Height (m) | Slope of the Riverbed (m/km) | Location |
|---|---|---|---|---|---|---|---|---|---|
| Yenisei | 0.25–2.6 | 2–3 | 3487 | 1020 | 102,806 | 650 | 1196 | 16.8 | Ulug-Khem basin |
| Big Yenisei | 1.4–2.4 | 1.5–4 | 605 | 594 | 57,766 | 630 | 1448 | 3.1 | Todzhin-skaya basin; Kyzyl basin |
| Small Yenisei | 1.8–2.3 | 1–2.4 | 563 | 411 | 36,395 | 636 | 1555 | 2.8 | Sangilen Highlands; Ulugh-Khem basin |
| Tes-Hem | 1.1–2.1 | 1–2.1 | 757 | 55.6 | 18,430 | 1067 | 1842 | 7.9 | East Tannu-Ola |
| Hem-chik | - | 0.75–2 | 320 | 102 | 3268 | 850 | 1923 | 14.4 | Shapshal ridge; Khemchik basin |
| Alash | 0.43 | 0.30–2 | 172 | | 4741 | 920 | 2063 | 9.2 | Alash Plateau |
| Ak—Sug | 0.31 | 0.25–1 | 160 | 14 | 997.4 | 1150 | 1966 | 26.8 | Alash Plateau |
| Chadan | - | - | 98 | - | 881.5 | 800 | 1567 | 28.8 | West Tannu-Ola |
| Durgen | 0.54 | 0.66–1 | 93 | - | 121.7 | 1200 | 1751 | 42.3 | The northern slope of East Tannu-Ola |
| Chaa-Hol | 0.28 | 0.5–2 | 90 | - | 320.3 | 540 | 1694 | 43.1 | The northern slope of the western Tannu-Ola |
| Huule (Torgalyg) | - | 0.4–2 | 53 | - | 1090 | 535 | 1273 | 30.9 | The northern slope of the Eastern Tannu-Ola; The Central Tuva basin |
| Anyyak-Chyrgaki | 0.173 | 0.2–2 | 52 | - | 1859 | 800 | 1519 | 13.9 | West Tannu-Ola |
| Dyttyg-Hem | - | 0.2–0.8 | 34 | - | 426.9 | 1250 | 1710 | 36.3 | Southern slope of East Tannu-Ola |
| Biche-Bayan-Kol | 0.34 | 0.3–0.8 | 32 | - | 15.3 | 750 | 1222 | 26.0 | Uyuk Ridge |
| Adyr-khem | 0.17 | 0.5–2 | 8.25 | - | 8.25 | 1850 | 2076 | 66.2 | Alash Plateau |

**Table 2.** General information about the studied lakes [15].

| Name | Depth (m) | Water Mirror Area, $km^2$ | Type | Height (m) | Location |
|---|---|---|---|---|---|
| **Tore-Khol** | 6–8 (max. 40 m) | 68.8 | Freshwater | 1148 | Ubsunur basin |
| **Chagytai** | 17 | 28.6 | Freshwater lake | 1005 | The foot of the northern slope of the Tannu-Ola ridge |
| **Cheder** | 1.5–2 | 4.3 | Salt lake | 706 | South of the Tuva basin, a drainless depression |
| **Thermokarst 1** | 4 | 0.3 | Thermokarst lake | 1850 | Alash Plateau |
| **Thermokarst 2** | 5 | 0.1 | Thermokarst lake | 1850 | Alash Plateau |

## 2.2. Analytical Methods

The list of measured parameters included temperature, pH, electrical conductivity (E.C.) and concentration of dissolved gases ($CO_2$ and $O_2$), dissolved organic (DOC) and inorganic carbon (DIC), optical properties of organic matter, as well as $CO_2$ emission flux from the water surface. Dissolved oxygen, pH, electrical conductivity and temperature were measured in situ using an EXO2 multiparameter probe and a WTW Multi 3320 multimeter. The measurement of $pCO_2$ in water was carried out in situ using the GM70 data logger Vaisala®. The $pCO_2$ was measured in situ by an infrared gas analyzer (IRGA, GMT222, Vaisala, Finland) [70]. The sensor was enclosed in a semipermeable membrane and placed

directly into the surface water (30–50 cm depth), where it was allowed to equilibrate for approximately 30 min. The sensors were calibrated against standard gas mixtures (0, 800, 3000, 8000 ppm) before and after the sampling. Following calibration, results were corrected using water temperature and barometric pressure.

Carbon dioxide emissions from the water surface were measured by direct floating chamber method using SensAir sensors. We used a freely drifting chamber (30 cm diameter, covered with aluminum tape). The $CO_2$ accumulation rate inside the chamber was recorded continuously at 5 sec intervals for 5–10 min and used to compute (by linear regression if $R^2 > 0.75$) $CO_2$ flux and $k_{CO_2}$ following Kuhn et al., 2018 [71]. For all calculations, the $CO_2$ air–water equilibrium was calculated assuming an air concentration of 400 ppm. Further details of $pCO_2$ and $fCO_2$ measurements in rivers and lakes of adjacent regions are provided elsewhere [1,72–75].

River and lake water was collected from the surface (depth 0.5 m) in a precleaned polypropylene container with a capacity of 1 L and immediately filtered through nitrate-cellulose filters (<0.45 microns, Sartorius Minisart High Flow). DOC and DIC were measured in the BIO-GEO-CLIM Laboratory (TSU) using a total organic carbon analyzer of the TOC-LCSN series, Shimadzu, with an uncertainty of 2%. As an indicator of the quality of DOM, we measured visual and UV absorbance using spectrophotometry (Agilent Cary 300 spectrophotometer). These measurements allowed for the assessment of spectral slopes ($S_{290–350}$, $S_{275–295}$, $S_{350–400}$); the slope ratio ($S_R$) is useful for approximating the molecular weight of DOM and the degree of its aromaticity. The relative amount of aromatic compounds was assessed via specific UV absorbance ($SUVA_{254}$). The weight-average molecular weight index (WAMW) of DOM, which reflects the degree of polymerization, was quantified using the absorbance at 280 nm. Finally, the humification index was characterized using the ratio of absorbance at 250 and 400 nm (the E2:E4 ratio).

Total bacterial cell (TBC) concentration was measured after sample fixation in glutaraldehyde (in the field immediately after collection) using flow cytometry (Guava® EasyCyteTM systems, Merck, Rahway, NJ, USA). Cells were stained using 1 μL of 10 times diluted SYBR GREEN solution (10,000×, Merck), added to 250 μL of each sample before analysis. Particles were identified as cells based on green fluorescence and forward scatter.

To build the maps, the DAICHI satellite (ALOS) survey with a resolution of 30 $m^2$ was used together with the 3D Analysis module in the ArcGIS environment. To delineate the catchments of the studied rivers, the Flow Direction module was used in the ArcGIS environment, which calculates flow lines based on data on the heights of nearby points.

Carbon concentrations and fluxes measured in rivers and lakes were tested for normality using a Shapiro–Wilk test. In case the data were not normally distributed, we used nonparametric statistics. Comparisons of dissolved C parameters between seasons and groups of the water bodies were conducted using a nonparametric Mann–Whitney test at a significance level of 0.05. For comparison of unpaired data, a nonparametric H-criterion Kruskal–Wallis test was used to assess the differences between sites. The Pearson rank order correlation coefficient ($p < 0.05$) was used to determine the relationship between $CO_2$ concentrations and flux and the main hydrochemical parameters of the water column, which are considered as potential drivers (pH, $O_2$, water temperature, electrical conductivity, DOC, DIC and total bacterial number). Boxplots of median, IQR and outlies were constructed for visualization of the differences between different seasons and groups of rivers or lakes.

## 3. Results

### 3.1. Major Hydrochemical Parameters

The values of pH varied significantly during different seasons of the year (F = 11.31, $p = 0.000$, Figure 3a) and between the two major types of the water bodies—lakes and rivers (F = 10.67, $p = 0.000$; Figure 3b,c). The electrical conductivity (E.C., μS $cm^{-1}$) in the studied water bodies varied widely and increased in the following order: thermokarst lakes

$(29 \pm 22) <$ large rivers $(172 \pm 97) <$ small rivers $(207 \pm 121) <$ freshwater lakes $(534 \pm 252) <$ salt lake $(54,600 \pm 24,380)$.

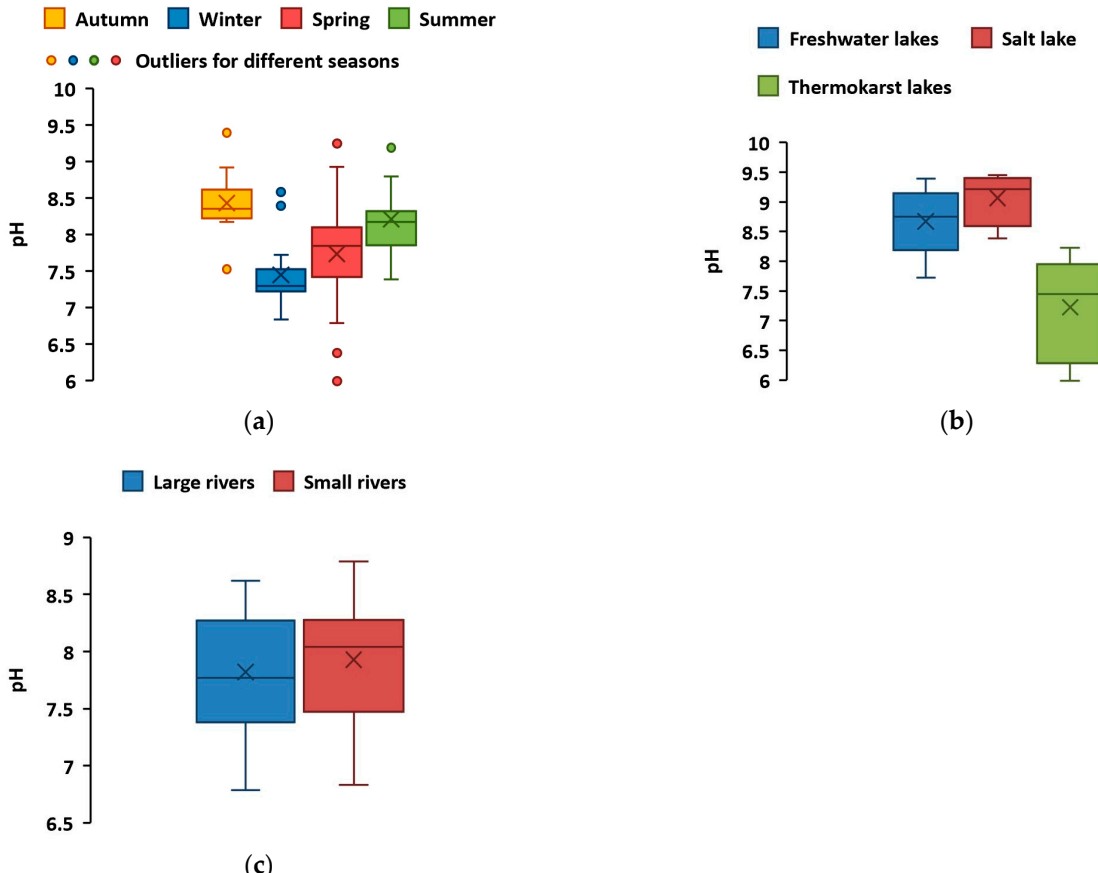

**Figure 3.** Boxplot of median pH values and IQR range (with outlies as dots) (**a**) during different seasons of the year (both rivers and lakes), and separately in (**b**) rivers and (**c**) lakes.

The water bodies of the Alash plateau, both rivers and lakes, exhibited the lowest E.C. In winter and autumn during low-water season, the electrical conductivity was significantly higher than in spring and summer (F = 3.3, $p = 0.03$), see Supplementary Figures S1 and S2. For lakes, a significant relationship ($p < 0.05$) between electrical conductivity and the pH was established (Figure 4). No such relationship was revealed for rivers. A strong direct relationship between electrical conductivity and DIC content was also observed (Figure 5), which reflected the dominance of bicarbonate ions in the major salt composition of both rivers and lakes (Supplementary Table S1).

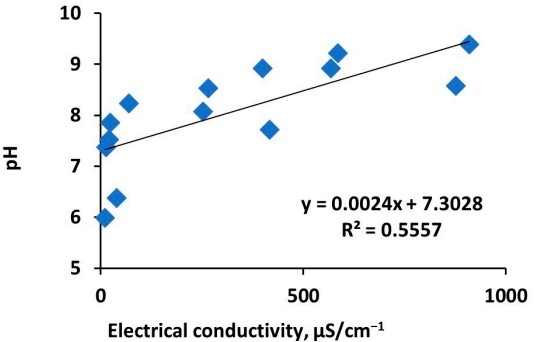

**Figure 4.** The relationship between electrical conductivity and DIC concentrations in the waters of the studied lakes (the salt lake Cheder is excluded).

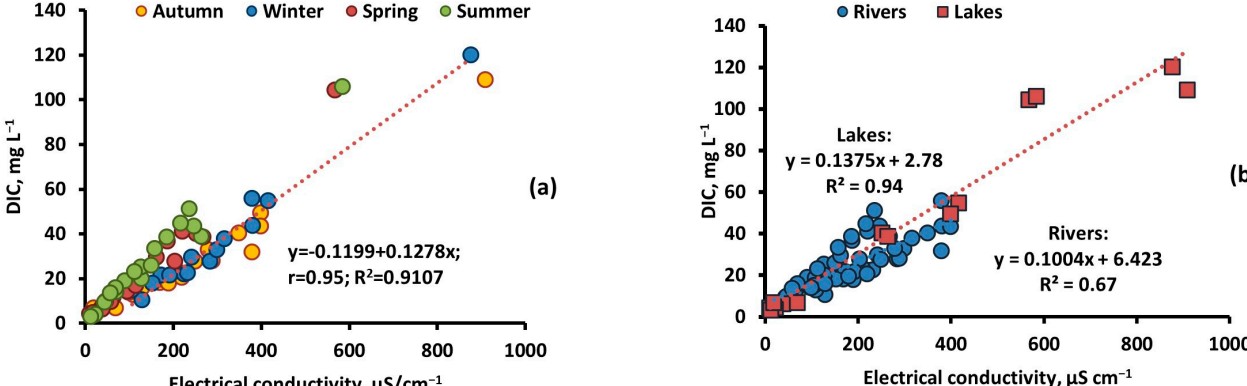

**Figure 5.** Linear relationship between DIC concentration and electrical conductivity in rivers and lakes of the Tyva region (**a**) in different seasons (lakes and rivers together) and (**b**) in different types of objects (averaged across seasons).

### 3.2. DIC vs. DOC Concentrations

Elevated DIC concentrations were observed in freshwater lakes, especially in Torehol Lake where they reached 120 mg $L^{-1}$, likely due to the impact of carbonate-rich groundwaters, which are reported to occur within the lake watershed (Figure 6a). The minimal DIC values were recorded in thermokarst lake waters, from 2.8 to 6.6 mg $L^{-1}$. The seasonal dynamics of DIC demonstrated rather low variations (within 30%) with minimal values observed in spring and maximal ones in winter, see Figure 6b.

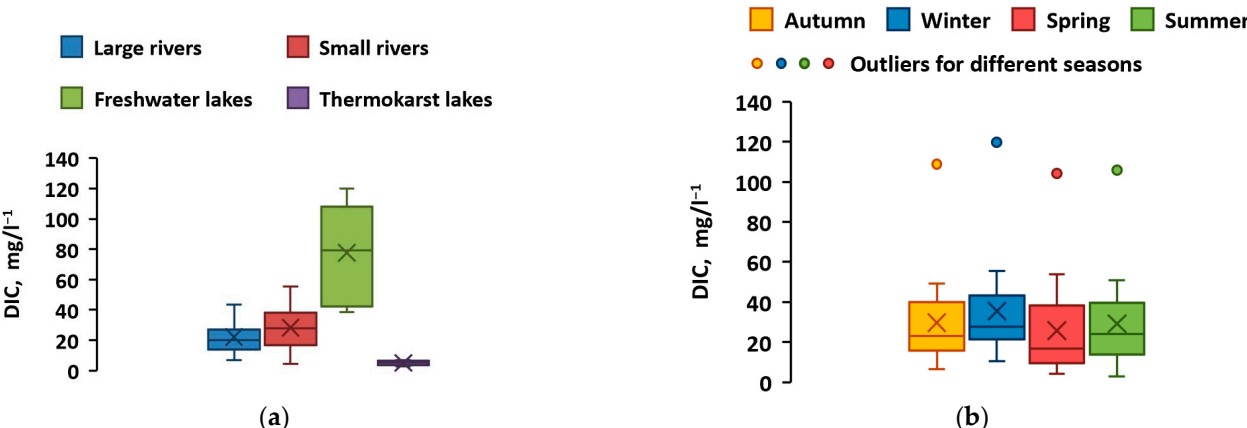

**Figure 6.** Median (and IQR) boxplots of DIC concentration (mg/$L^{-1}$) (**a**) in different types of objects (averaged across seasons) and (**b**) in different seasons (lakes and rivers together).

The maximal DOC values were measured in the waters of Torehol Lake and the thermokarst lakes. Of all the rivers studied, the Dyttyg-Khem, Durgen and Biche-Bayan-Kol waters were highly enriched in DOM, whereas the majority of the water bodies ranged from 2 to 6 mg $L^{-1}$ in DOC concentration. As in the case of DIC spatial and temporal patterns (Figure 6), the DOC concentrations demonstrated relative stability across seasons (Figure 7), with an exception of an anomalously high value (14.8 mg $L^{-1}$) in Torehol Lake during winter. The highest SUVA$_{254}$ values, which reflect the DOM aromaticity, were observed in the waters of thermokarst lakes, followed by rivers, whereas the minimal values were recorded in freshwater lakes (Figure 8a–c). The highest SUVA$_{254}$ was recorded in spring and the lowest in winter (Figure 8b).

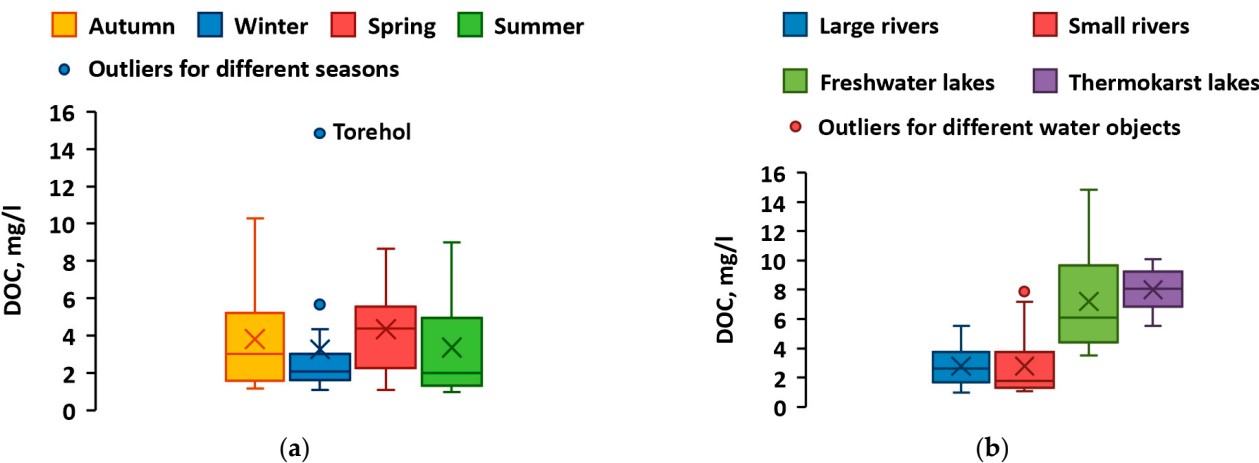

**Figure 7.** Median (and IQR) boxplots of DOC concentration (mg L$^{-1}$) in different types of objects (**a**) in different seasons (lakes and rivers together) and (**b**) in different types of objects (averaged across seasons).

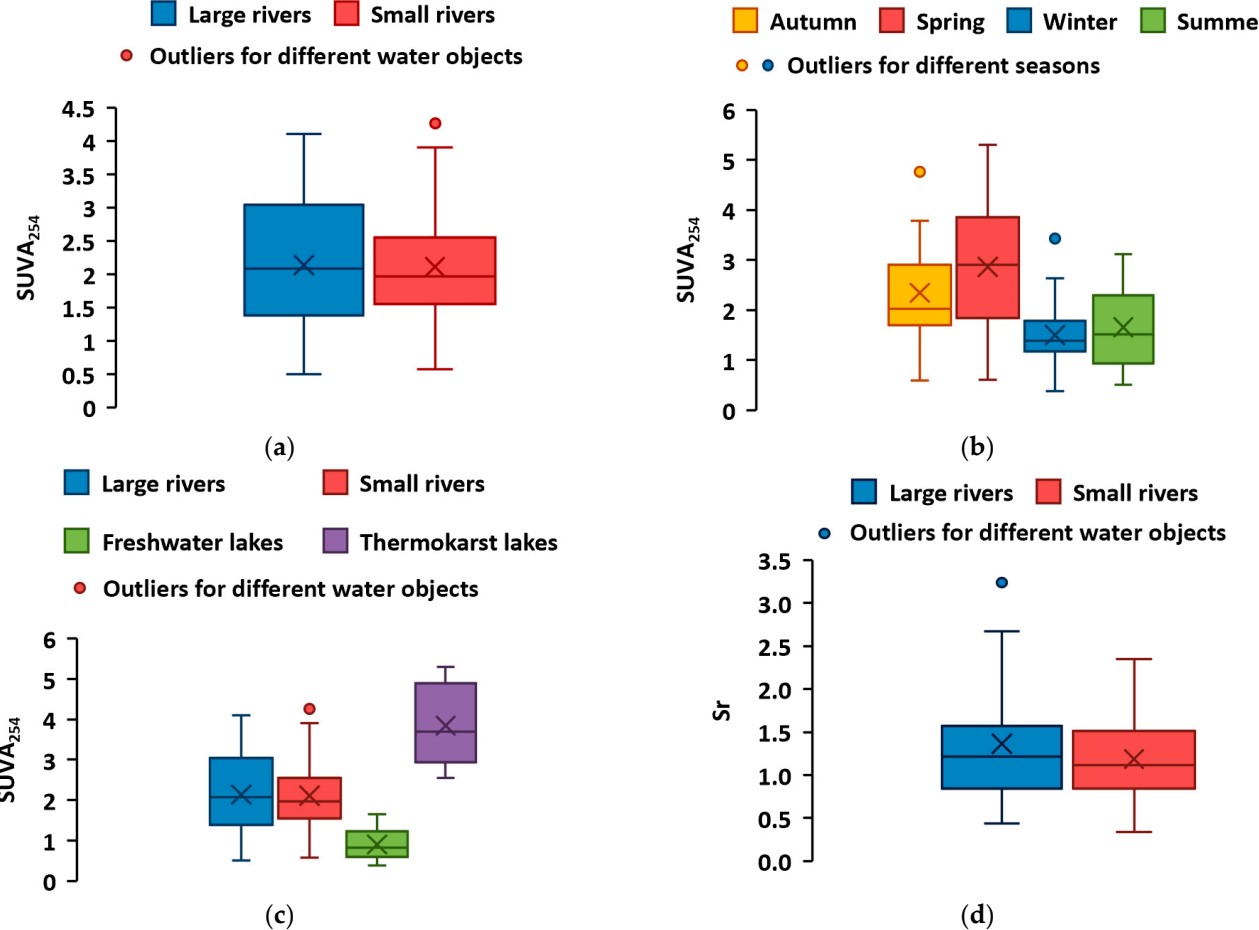

**Figure 8.** Median (and IQR) boxplots of SUVA$_{254}$ (**a**) in large and small rivers (averaged across seasons); (**b**) in different seasons (lakes and rivers together); (**c**) in different types of objects (averaged across seasons) and SR; and (**d**) in large and small rivers (averaged across seasons).

The ratio E$_{254}$:E$_{436}$ is known to indicate the relative role of allochthonous versus autochthonous organic substances in water bodies [76,77]. The DOM composition in the studied water objects was dominated by allochthonous substances, with the exception of the summer and part of the spring periods, when there was an increase in the E$_{254}$:E$_{436}$

ratio, indicating an active process of photosynthesis and destruction of detritus in the rivers and lakes themselves, leading to an increase in autochthonous organic matter [78,79].

There was no strong variation in the value of $SUVA_{254}$ in the waters of both small and large rivers (Figure 8a). However, small river waters had a lower $S_R$ value, which indicated an increased degree of aromaticity compared to large rivers (Figure 8d). A high index of $SUVA_{254}$ ($3.8 \pm 1$ L mg$^{-1}$ m$^{-1}$) and a low index of $S_R$ ($0.9 \pm 0.1$) in the waters of thermokarst lakes indicated the presence of high-molecular-weight aromatic compounds. In contrast, freshwater lakes, poor in DOC, exhibited much lower $SUVA_{254}$ and higher SR values compared to thermokarst lakes. The ratio of optical densities $E_{254}{:}E_{436}$ in rivers followed the order 'summer > spring > autumn > winter' (Figure 9a). This ratio was much higher in thermokarst lakes compared to rivers and freshwater lakes (Figure 9b).

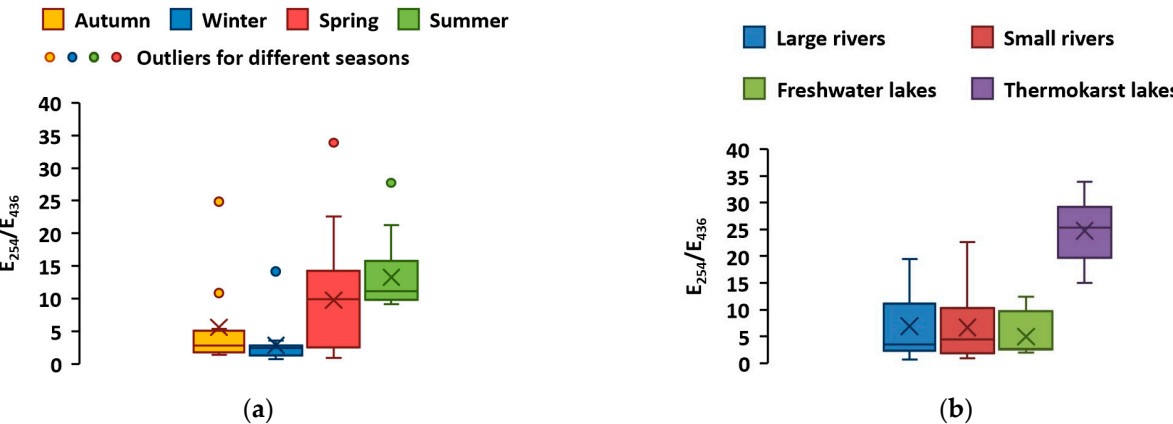

**Figure 9.** The ratio of optical densities $E_{254}/E_{436}$ in the rivers and lakes of Tyva (**a**) during different seasons, and (**b**) for different types of objects, averaged across seasons.

### 3.3. Spatial and Seasonal Pattern of $pCO_2$ and $fCO_2$ in Rivers and Lakes

The highest values of $pCO_2$ were noted for a group of small rivers, with a maximum in the Adyr-Khem River, the smallest of the studied rivers draining the mound peat bog (Table 3). A high content of dissolved $CO_2$ was also noted in the waters of the Chadan River, originating in the western Tannu-Ola, and in the waters of the Khule River (Torgalyg), whose source is located on the eastern Tannu-Ola. This region (Tannu-Ola) is known for its $CO_2$-rich underground discharges at the earth's surface. The average values of $pCO_2$ in large rivers differ significantly from small ones; the minimum was marked for the Alash River (Figure 10, Table 3).

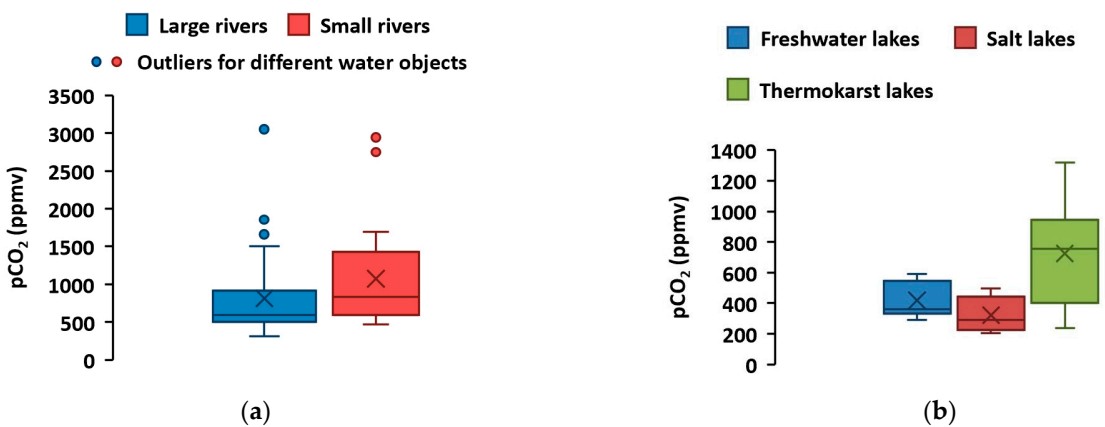

**Figure 10.** Median (and IQR) boxplots of $CO_2$ partial pressure (ppmv) in (**a**) rivers and (**b**) lakes, averaged across seasons. Different groups of rivers (F = 5.42, $p = 0.02$) and lakes (F = 4.44, $p = 0.03$) are significantly different.

**Table 3.** The values of $pCO_2$ (ppmv) in rivers and lakes of the region.

| Rivers | Average | Median |
|---|---|---|
| Large rivers | | |
| Yenisei | 689 | 578 |
| Big Yenisei | 860 | 587 |
| Small Yenisei | 790 | 571 |
| Tes-Khem | 672 | 609 |
| Khemchik | 705 | 715 |
| Alash | 563 | 502 |
| Small rivers | | |
| Ak—Sug | 809 | 552 |
| Chadan | 1470 | 1112 |
| Durgen | 739.5 | 493 |
| Chaa-Hol | 778 | 729 |
| Huule (Torgalyg) | 1133 | 1003 |
| Anyyak-Chyrgaki | 909 | 932 |
| Dyttyg-Hem | 840 | 660 |
| Biche-Bayan-Kol | 743 | 743 |
| Adyr-Khem | 2043 | 2105 |
| Lakes | | |
| Tore-Khol | 332 | 337 |
| Chagytai | 504 | 535 |
| Cheder | 321 | 292 |
| Thermokarst lake 1 | 754 | 705 |
| Thermokarst lake 2 | 694 | 806 |

In lakes, the dissolved $CO_2$ content was sizably lower than that in rivers; the minimum values were noted for the salt lake Cheder and Tore-Khol Lake, located within a sandy substrate.

In terms of seasonal variations, in both rivers and lakes, the $pCO_2$ was significantly higher in winter due to $CO_2$ accumulation under ice (Figure 11). Note that because shallow thermokarst lakes freeze solid in winter, measurements of the water column during this period in these water bodies were not possible.

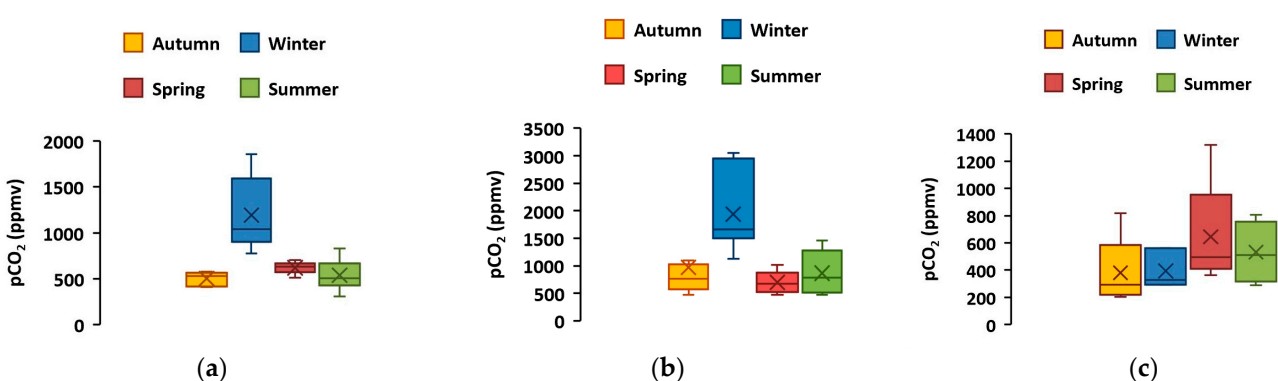

**Figure 11.** Median (and IQR) boxplots of $CO_2$ partial pressure (ppmv) in (**a**) large and (**b**) small rivers across seasons, which are statistically different (large rivers, F = 12.20, $p$ = 0.001; small rivers, F = 7.801, $p$ = 0.001, accordingly). (**c**) Median (and IQR) boxplots of $CO_2$ partial pressure (ppmv) in lakes across seasons (F = 0.9223, $p$ = 0.46).

Overall, $CO_2$ emissions from the surface of rivers and lakes were quite low (median 0.05–0.15 g C m$^{-2}$ d$^{-1}$; Table 4) and did not differ significantly ($p$ > 0.05) among water bodies, including the two main categories—rivers and lakes (Figure 12a,b). There was no

significant difference in $fCO_2$ between different seasons of the year (Figure 12c). Exceptionally high $CO_2$ fluxes in water were encountered in the Chadan River and the Ak-Sug River (12 and 5.6 g C m$^{-2}$ d$^{-1}$, respectively). Despite their small size, these rivers do not freeze solid during the ice-on period, and they likely possess sizable bicarbonate/$CO_2$-rich underground sources that are discharged at the river bed and hence are mostly pronounced during the winter baseflow.

**Table 4.** Measured $fCO_2$ fluxes (g C m$^{-2}$ d$^{-1}$) in rivers and lakes of the Tyva Republic (Sayan–Altai Mountain region).

| | Autumn | Winter | Spring | Summer | Average Value | Median | Standard Deviation |
|---|---|---|---|---|---|---|---|
| Large rivers | | | | | | | |
| Yenisei | 0.10 | 0.39 | 0.06 | 0.08 | 0.16 | 0.09 | 0.15 |
| Big Yenisei | | 0.22 | 0.11 | 0.09 | 0.14 | 0.11 | 0.07 |
| Small Yenisei | 0.19 | 0.71 | 0.03 | 0.12 | 0.26 | 0.15 | 0.31 |
| Tes-Khem | 0.42 | 0.16 | 0.10 | 0.10 | 0.19 | 0.13 | 0.15 |
| Khemchik | 0.19 | 0.35 | 0.00 | 0.08 | 0.16 | 0.14 | 0.15 |
| Alash | 0.07 | 0.10 | 0.00 | −0.02 | 0.04 | 0.03 | 0.06 |
| Small rivers | | | | | | | |
| Ak—Sug | 0.10 | 5.60 | 0.04 | 0.11 | 1.46 | 0.10 | 2.76 |
| Chadan | | 12.11 | 0.01 | 0.13 | 4.08 | 0.13 | 6.95 |
| Durgen | 0.20 | 0.25 | 0.00 | 0.02 | 0.12 | 0.11 | 0.13 |
| Chaa-Hol | 0.31 | | 0.04 | 0.15 | 0.17 | 0.15 | 0.14 |
| Huule (Torgalyg) | 0.14 | 0.15 | 0.05 | 0.45 | 0.20 | 0.15 | 0.17 |
| Anyyak-Chyrgaki | 1.89 | | 0.06 | 0.33 | 0.76 | 0.33 | 0.99 |
| Dyttyg-Hem | | 0.51 | 0.12 | 0.03 | 0.22 | 0.12 | 0.26 |
| Biche-Bayan-Kol | | | | | | | |
| Adyr-Khem | 3.58 | 0.48 | 0.16 | −0.06 | 1.04 | 0.32 | 1.71 |
| Lakes | | | | | | | |
| Tore-Khol | 0.03 | 0.02 | 0.07 | 0.11 | 0.06 | 0.05 | 0.04 |
| Chagytai | 0.16 | 0.01 | 0.11 | 0.00 | 0.07 | 0.06 | 0.08 |
| Cheder | 0.07 | 0.09 | 0.05 | −0.04 | 0.04 | 0.06 | 0.06 |
| Thermokarst lake 1 | 0.09 | | 0.06 | 0.04 | 0.06 | 0.06 | 0.02 |
| Thermokarst lake 2 | 0.13 | | 0.02 | 0.06 | 0.07 | 0.06 | 0.06 |

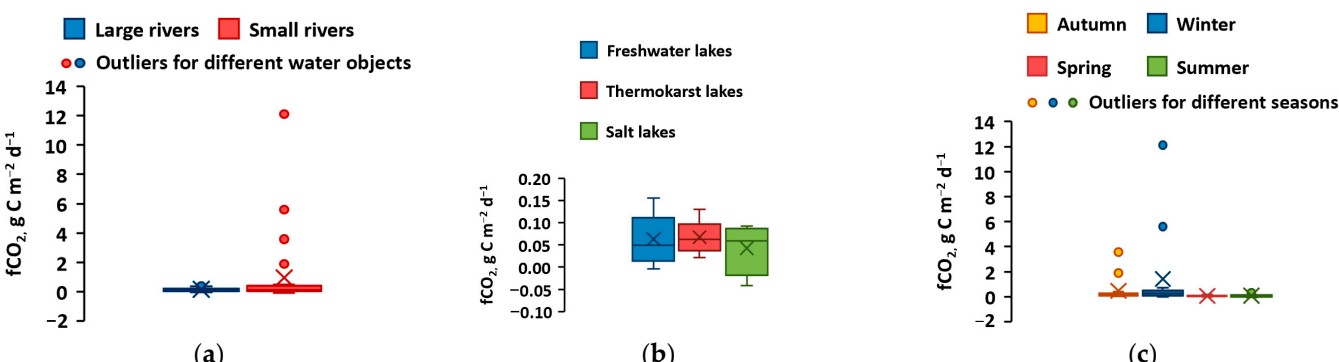

**Figure 12.** Boxplots (median and IQR range) of $fCO_2$ in (**a**) rivers, where the difference is insignificant (F = 2.30, $p$ = 0.13); (**b**) lakes (F = 0.30, $p$ = 0.743); and (**c**) all types of objects across seasons (F = 2.54, $p$ = 0.064).

Analysis of correlations between $O_2$ and $CO_2$ concentrations of all objects across seasons did not allow us to establish any relationship (r = −0.13, $p$ > 0.05); however, when

considering solely large rivers and thermokarst lakes, a negative relationship (r = −0.47 and r = −0.95) was observed (Table S1; Figure S4). In freshwater lakes, $pCO_2$ also negatively correlated with $O_2$ level (r = −0.50, $p < 0.05$). The water temperature exerted a significant (r = 0.64, $p < 0.05$) impact on $pCO_2$ in large rivers (Table S1).

*3.4. Testing Potential Drivers of $CO_2$ Concentrations and Fluxes*

During the summer period, the $pCO_2$ of the water column in lakes correlated with the lake water surface area ($p < 0.05$), whereas during other seasons, this relationship was not significant ($p < 0.05$); see Figure 13a. The flux of $CO_2$ positively correlated with lake surface area during the spring and winter periods ($p < 0.01$), as illustrated in Figure 13b. In thermokarst lakes, $pCO_2$ and $fCO_2$ increased with lake surface area in spring but decreased with $S_{area}$ in summer and autumn (Figure S5), although these qualitative trends could not be statistically supported due to too low a number of sampled lakes. The river watershed area positively correlated with $pCO_2$ and $fCO_2$ ($p < 0.05$) in the autumn period, whereas during other seasons, the correlation was absent ($p > 0.05$), as shown in Figure S6.

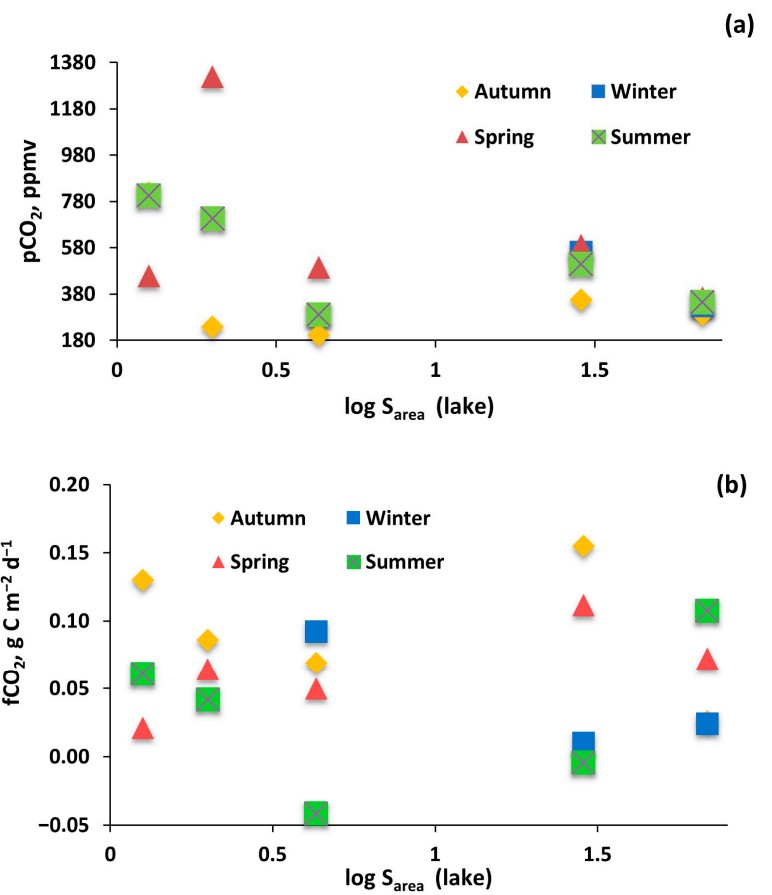

**Figure 13.** The relationship between (**a**) $pCO_2$ and (**b**) $fCO_2$ and lake water surface area of the Tyva region (all seasons together).

Among the 'internal' factors likely to control the $CO_2$ pattern (concentration and fluxes) in the water bodies, we tested major hydrochemical parameters of the water column, including pH, conductivity, DIC, DOC, concentration, DOM quality and a microbiological parameters (i.e., total bacterial count), as presented in Table 5. Considering all water objects together, neither $pCO_2$ nor $fCO_2$ correlated significantly (r > 0.39 at $p = 0.05$) with any tested parameter. Note that this result does not contradict several significant relationships of $pCO_2$ with lake area or $O_2$, which were analyzed for individual seasons. Individual analyses of groups of rivers and lakes demonstrated strong positive control ($p < 0.05$) of

E.C. and pH on fCO$_2$ in large rivers (r = 0.5 and 0.64, respectively) and negative control of pH on pCO$_2$ in lakes (r = −0.99); see Table S1 of the Supplement.

**Table 5.** Pearson correlation coefficients between major hydrochemical and microbiological parameters of the water column, including pH, O$_2$, S.C., DOC, DIC and optical parameters of DOM reflecting DOM quality. S$_{area}$, E.C. and TBC represent the area of the lake surface or the river watershed, electrical conductivityf and total bacterial count, respectively.

| All Objects | E.C. | T$_{water}$ | pH | O$_2$ | pCO$_2$ | TBC | DIC | DOC | SUVA$_{254}$ | E2:E3 | E$_{254}$:E$_{436}$ | S$_R$ | fCO$_2$ |
|---|---|---|---|---|---|---|---|---|---|---|---|---|---|
| **All** | **E.C.** | **T$_{water}$** | **pH** | **O$_2$** | **pCO$_2$** | **TBC** | **DIC** | **DOC** | **SUVA$_{254}$** | **E2:E3** | **E$_{254}$:E$_{436}$** | **S$_R$** | **fCO$_2$** |
| S$_{area}$ | 0.03 | 0.16 | 0.23 | 0.23 | −0.18 | −0.29 | −0.07 | −0.46 | −0.2 | −0.05 | −0.41 | −0.14 | −0.16 |
| E.C. | | 0.05 | 0.76 | 0.25 | −0.24 | −0.01 | 0.95 | 0.23 | −0.75 | 0.72 | −0.48 | 0.85 | −0.06 |
| T$_{water}$ | | | −0.26 | −0.69 | −0.12 | 0.02 | 0.19 | 0.11 | 0.01 | −0.37 | 0.12 | −0.11 | −0.33 |
| pH | | | | 0.54 | −0.33 | 0.09 | 0.70 | −0.01 | −0.77 | 0.71 | −0.58 | 0.61 | −0.04 |
| O$_2$ | | | | | −0.13 | −0.16 | 0.06 | −0.28 | −0.42 | 0.53 | −0.46 | 0.25 | 0.09 |
| pCO$_2$ | | | | | | 0.00 | −0.24 | −0.28 | 0.29 | −0.07 | 0.01 | −0.15 | 0.85 |
| TBC | | | | | | | 0.14 | 0.33 | 0.09 | −0.03 | 0.22 | −0.16 | 0.05 |
| DIC | | | | | | | | 0.26 | −0.72 | 0.64 | −0.44 | 0.75 | −0.08 |
| DOC | | | | | | | | | 0.25 | 0.03 | 0.69 | 0.21 | −0.23 |
| SUVA$_{254}$ | | | | | | | | | | −0.61 | 0.81 | −0.56 | 0.08 |
| E2:E3 | | | | | | | | | | | −0.51 | 0.75 | 0.21 |
| E$_{254}$:E$_{436}$ | | | | | | | | | | | | −0.39 | −0.16 |
| S$_R$ | | | | | | | | | | | | | 0.06 |
| **Rivers** | **E.C.** | **T$_{water}$** | **pH** | **O$_2$** | **pCO$_2$** | **TBC** | **DIC** | **DOC** | **SUVA$_{254}$** | **E2:E3** | **E$_{254}$:E$_{436}$** | **S$_R$** | **fCO$_2$** |
| S$_{area}$ | 0.22 | 0.38 | 0.17 | 0.02 | −0.46 | −0.23 | 0.18 | 0.08 | −0.06 | −0.04 | 0.05 | −0.06 | −0.40 |
| E.C. | | −0.16 | 0.79 | 0.54 | −0.05 | −0.11 | 0.88 | −0.21 | −0.77 | 0.42 | −0.53 | 0.34 | −0.04 |
| T$_{water}$ | | | −0.37 | −0.71 | −0.19 | 0.16 | 0.06 | 0.11 | 0.18 | −0.72 | 0.28 | −0.46 | −0.36 |
| pH | | | | 0.67 | −0.23 | −0.18 | 0.65 | −0.18 | −0.78 | 0.61 | −0.55 | 0.46 | −0.09 |
| O$_2$ | | | | | −0.07 | −0.26 | 0.16 | −0.16 | −0.48 | 0.74 | −0.44 | 0.66 | 0.06 |
| pCO$_2$ | | | | | | 0.22 | −0.05 | −0.30 | 0.16 | 0.17 | −0.20 | 0.17 | 0.93 |
| TBC | | | | | | | 0.02 | 0.32 | 0.17 | −0.17 | 0.28 | −0.40 | 0.12 |
| DIC | | | | | | | | −0.11 | −0.68 | 0.14 | −0.39 | 0.08 | −0.08 |
| DOC | | | | | | | | | 0.45 | −0.27 | 0.89 | −0.54 | −0.24 |
| SUVA$_{254}$ | | | | | | | | | | −0.36 | 0.75 | −0.39 | 0.11 |
| E2:E3 | | | | | | | | | | | −0.49 | 0.65 | 0.33 |
| E$_{254}$:E$_{436}$ | | | | | | | | | | | | −0.58 | −0.20 |
| S$_R$ | | | | | | | | | | | | | 0.28 |
| **Lakes** | **E.C.** | **T$_{water}$** | **pH** | **O$_2$** | **pCO$_2$** | **TBC** | **DIC** | **DOC** | **SUVA$_{254}$** | **E2:E3** | **E$_{254}$:E$_{436}$** | **S$_R$** | **fCO$_2$** |
| S$_{area}$ | 0.95 | 0.58 | 0.72 | 0.11 | −0.31 | 0.19 | 0.92 | −0.32 | −0.93 | −0.53 | −0.83 | 0.98 | 0.17 |
| E.C. | | 0.56 | 0.75 | 0.17 | −0.38 | 0.41 | 0.99 | −0.15 | −0.87 | −0.25 | −0.77 | 0.92 | 0.07 |
| T$_{water}$ | | | −0.07 | −0.51 | 0.19 | −0.38 | 0.57 | −0.29 | −0.45 | −0.14 | −0.41 | 0.46 | −0.53 |
| pH | | | | 0.59 | −0.54 | 0.65 | 0.71 | 0.16 | −0.68 | −0.28 | −0.53 | 0.75 | 0.55 |
| O$_2$ | | | | | −0.92 | 0.54 | 0.14 | 0.20 | −0.35 | −0.03 | −0.30 | 0.18 | 0.16 |
| pCO$_2$ | | | | | | −0.46 | −0.35 | 0.05 | 0.59 | 0.08 | 0.57 | −0.36 | 0.10 |
| TBC | | | | | | | 0.45 | 0.34 | −0.19 | 0.39 | −0.16 | 0.26 | 0.36 |
| DIC | | | | | | | | −0.17 | −0.84 | −0.19 | −0.76 | 0.90 | 0.05 |
| DOC | | | | | | | | | 0.48 | 0.68 | 0.67 | −0.42 | 0.02 |
| SUVA$_{254}$ | | | | | | | | | | 0.58 | 0.96 | −0.94 | −0.06 |
| E2:E3 | | | | | | | | | | | 0.59 | −0.59 | −0.42 |
| E$_{254}$:E$_{436}$ | | | | | | | | | | | | −0.89 | −0.02 |
| S$_R$ | | | | | | | | | | | | | 0.28 |

We found that TBC does not correlate with any parameter of the CO$_2$ system in rivers except in weak positive correlations with DOC (R$_{Pearson}$ = 0.32, $p > 0.05$). In lakes, there were weak but not significant ($p > 0.05$) correlations with DOC (R$_{Pearson}$ = 0.31) and fCO$_2$ (R$_{Pearson}$ = 0.36, Table 5). This mostly occurred in thermokarst lakes (Table S1). It is therefore possible that some bacterial processing of DOC in the water column (biodegradation) generates excessive CO$_2$ emissions. However, these processes are obscured by other external factors of CO$_2$ delivery to the river or lake, which occur throughout soil and surface waters, respiration of sediments or biodegradation of particulate organic matter.

It is also important to note that wind speed might strongly affect the CO$_2$ emission fluxes, especially in wind-unprotected large lakes. Unfortunately, because we did not use wind-speed-based flux calculations [75] in this study, some environmental controllers listed in Table 5 could have exhibited a lower impact on fCO$_2$ than it occurs under field conditions.

## 4. Discussion

### 4.1. Major Solutes, Dissolved Organic and Inorganic Carbon

It is known that inland water systems are sizable sources of $CO_2$, which is a highly important prerequisite for the retroactive link between the climate and C cycle in aquatic systems [80–82]. Although at present, the climate changes in the territory of Tyva are already strongly pronounced (e.g., ref. [83]), it is not yet possible to quantify this impact on water bodies. However, the present study can provide a solid background for assessing the current status of C cycle in order to be able to judge the future changes which are likely to occur over the next decades.

The level of DIC concentration was sizably higher than that of DOC, and hence the C export in the studied rivers and C storage in the lakes are dominated by inorganic carbon. Only during spring was the DOC concentration higher than DIC, which could be linked to the snow melt, dilution of the groundwater signal and the dominance of surface flow. In particular, enhanced DOC delivery to lakes and rivers occurs during melt-snow interaction with surface soil and organic-rich litter. This period contrasts with the shallow subsurface flow in summer and autumn, when the water hydrochemical signal is essentially controlled by rock weathering at the watershed. In this regard, the present study corroborates the knowledge of processes controlling river and lake water chemical composition across seasons as established in the boreal, high latitude and subarctic rivers [84–86] and subarctic lakes [87]. Note here that the connection between the groundwaters and the lake, especially in permafrost-covered regions such the Altai–Sayan Mountain system, is weaker than that for rivers. A comobilization of DIC and other ions from shallow subsurface/groundwaters, bearing the signal of chemical weathering, for all water objects is confirmed by the observation that the DIC values strongly correlated with the electrical conductivity index. In the Khemchik, Ak-Sug, Alash and Adyr-Khem mountain rivers flowing in the northwestern part of the Republic of Tyva in the Western Sayans, the DIC concentration ranged from 4 to 20 mg $L^{-1}$ and in other rivers from 9~13 to 55 mg $L^{-1}$. It is possible that a reason for such reduced DIC values in mountain rivers is an increased precipitation in this area [83], the presence of permafrost [88] and high runoff [89]. An opposite situation is observed in the water bodies of the more arid Ubsu-Nur basin: Tore-Khol Lake and the Tes-Khem River, where the DIC values are the highest. Here, evaporative concentration of major solutes, occurring both within the river watersheds and the lake water column, can be responsible for elevated DIC and S.C. values.

Increased DOC concentrations were observed in lakes compared to rivers, which may be explained by higher water residence time and autochthonous production of OM. This was especially pronounced in thermokarst lakes, which exhibited the highest SUVA (Figure 8b,c) and humification index ($E_{254}/E_{436}$ ratio) (Figure 9b). The delivery of allochthonous material to the water bodies occurs essentially during summer and spring following the shallow water paths through organic-rich soil layers (Figure 9a). According to the ratio of optical densities $E_{254}/E_{436}$, allochthonous (peat-originated) organic matter prevails in thermokarst lakes and significantly exceeds the values of the ratios in other water bodies. Low water residence time in thermokarst lakes, their small size and their location within wetland zones may lead to sizable accumulation of terrestrial humic-rich organic matter. Much lower values of the optical index $E_{254}/E_{436}$ in rivers and freshwater lakes likely correspond to a decrease in the humification index [90] due to biotically driven degradation of humiclike aromatic components in these water objects with high water residence time [91].

### 4.2. Dissolved C Pattern and $CO_2$ Fluxes: Driving Factors and Comparison with Other Regions

We identified the oxygen regime as one of the main controlling factors of $pCO_2$ level in both rivers and lakes across seasons. The highest $CO_2$ concentrations were observed in $O_2$-impoverished waters, likely due to the important impact of partially anoxic sediment respiration on the $CO_2$ regime of the water column. Similar effects are reported in many boreal and subarctic waters across the world, notably in lakes and channels of the Ob River floodplain (e.g., ref. [73]).

The $pCO_2$ was generally higher in lakes of a smaller surface area, which is a well-known phenomenon in various lakes of the boreal and subarctic zones [1,75]. This can be linked to enhanced terrestrial input of $CO_2$-rich waters from the watershed in small lakes and sizable primary productivity of plankton and periphyton/macrophytes in large, mature lakes, leading to $CO_2$ uptake in the latter, as it is known in thermokarst water bodies in the neighboring territory of permafrost peatlands [92].

The seasonal variations of $CO_2$ concentration were characterized by a maximum $pCO_2$ during winter in rivers and during spring in lakes. The former can be explained by enhanced underground discharge within the river's main stem and $CO_2$ accumulation under ice, whereas the spring maximum of $CO_2$ in lakes is likely due to a lack of phytoplankton activity and the enhanced input of biologically labile terrestrial DOM from the watershed during freshet. This DOM can be subjected to intensive bacterial processing leading to a maximum of $CO_2$ emissions [93]. In the summer period, lakes act as a $CO_2$ sink due to biological production processes (macrophytes, cyanobacterial bloom) in the water column (e.g., ref. [94]).

The data on carbon (DOC, DIC, $CO_2$) concentrations and $CO_2$ fluxes in the Altai–Sayan mountain system were compared with the data reported for the Qinghai–Tibetan–Plateau [18,95,96], since this region has relatively similar climate and is also impacted by permafrost. To account for thermokarst lakes, we chose the largest compilation of lakes in the sporadic to continuous permafrost zone of Western Siberia, reported in Serikova et al. (2019) [1]. A synthesis of available information is provided in Table 6. The DOC levels of rivers and lakes of the Tuva Republic obtained in this study are quite similar within the range of natural seasonal and spatial variations to those in thermokarst lakes of the QTP as well as the Tibetan rivers. The DIC concentrations in lakes and rivers of the Sayan–Altai Mountain system are also similar to the values of the Qinghai–Tibetan plateau, with the exception of saline lakes, whose C cycle is strongly controlled by local evaporative processes.

**Table 6.** Comparison of carbon concentrations in lakes and rivers and $CO_2$ (mean and standard deviation) emissions from water surfaces in the Tyva region to other regions of the world that have similar climate and landscape parameters.

| Sites | Period | $pCO_2$ (ppmv) | $fCO_2$ (g C m$^{-2}$ d$^{-1}$) | DOC (mg L$^{-1}$) | DIC (mg L$^{-1}$) | Reference |
|---|---|---|---|---|---|---|
| All | Annual cycle | $1495 \pm 577$ | $0.46 \pm 1.6$ | $3.7 \pm 2.9$ | $29 \pm 24$ | This study |
| Rivers | Annual cycle | $929 \pm 611$ | $0.60 \pm 1.9$ | $2.7 \pm 1.7$ | $24.6 \pm 12$ | This study |
| Lakes | Annual cycle | $385 \pm 123$ | $0.06 \pm 0.05$ | $7.2 \pm 3.8$ | $78 \pm 35$ | This study |
| Thermokarst lakes | Annual cycle | $724 \pm 368$ | $0.067 \pm 0.038$ | $8 \pm 1.6$ | $5 \pm 1.6$ | This study |
| Fenghuoshan catchment, China | Annual cycle | $1260 \pm 145$ | $6.3 \pm 0.9$ | $17.9 \pm 5.5$ | $33 \pm 3$ | [19] |
| Thermokarst lakes, China | June–September | | | $9.5 \pm 5.7$ | $38 \pm 35$ | [96] |
| The NamCo basin and source area of Yellow River (SAID) | July 2015 (melting of glacier) | | | $2.3 \pm 1.3$ | $11.3 \pm 10.6$ | [97] |
| Yangtze River source region | Biweekly from May to October | $1086 \pm 275$ | | $2.8 \pm 0.6$ | $24.4 \pm 1.9$ | [18] |
| Qinghai Lake and the inflowing rivers | 23 May 2021 | | | $1.0 \pm 0.6$ | $13.3 \pm 7.7$ | [21] |
| Three catchments within the Nam Co watershed | June/July 2018, May 2019 and September 2019 | | | $2.4 \pm 0.3$ | $8.8 \pm 5.5$ | [98] |
| 76 lakes, Western Siberia Lowland | May–June, August–October | $1044 \pm 1540$ | $1.7 \pm 1.7$ | $16 \pm 10$ | $0.7 \pm 0.8$ | [1] |
| Lena upstream of Kirenga | 29 May to 17 June 2016 | $714 \pm 22$ | $0.85 \pm 0.06$ | $13.9 \pm 1.4$ | $20.0 \pm 1.2$ | [74] |
| Southwestern and northeastern regions of the QTP (70 lakes, four rivers and one reservoir on the QTP) | 20-year period (i.e., from the 2000s to the 2020s) | | $0.3 \pm 0.2$ | | | [95] |
| Two saline lakes (Qinghai Lake and Hala Lake) in the Tibetan Plateau | Continuously measured on 20, 23 October 2018 | | $13.1 \pm 0.4$ | | | [20] |

In contrast to the DIC and DOC concentrations, for which the results of different regions are fairly comparable, the $pCO_2$ in the Sayan–Altai Mountain region had generally lower values than those observed in the Qinghai–Tibetan Plateau and Western Siberia. It is possible that sizable autochthonous production and $CO_2$ uptake by plankton in lakes and by macrophytes in rivers could be partially responsible for lower $CO_2$ levels in the water bodies of the studied region during the summer period, when the water temperature becomes higher than that in subarctic Western Siberia or high-altitude QTP. At the same time, our results on $CO_2$ flux are comparable to those of Jia et al. (2022) [95] for the lakes and rivers of southwestern and northeastern regions of the QTP ($0.3 \pm 0.2$ (g C m$^{-2}$ d$^{-1}$)). However, unlike the maximal flux observed in the present work during winter, the latter authors reported the lowest flux during ice-covered seasons. The lowest $CO_2$ emission is typically observed during cold periods, as also reported by other studies [96–100]. In our case, high wintertime $pCO_2$ and $fCO_2$ could indicate the existence of sizable underground sources of $CO_2$ (discharge of $CO_2$-rich fluids, especially pronounced during low-water-level periods). These effects are visible for large rivers, which demonstrated strong positive impact of E.C., pH and DIC on $fCO_2$ (Table S1).

Overall, the differences of $CO_2$ emissions among different works on northern and mountain aquatic systems may be due to the peculiarities inherent to our region as well as differences in the methodologies used. It is interesting that the carbon uptake flux in the terrestrial Tyva steppe regions is estimated as $184 \pm 41$ g C m$^{-2}$ yr$^{-1}$ [101]. This is fairly comparable with the $CO_2$ emission flux from the rivers of the region, assessed in the present study (213 g C m$^{-2}$ yr$^{-1}$). Such a comparison is consistent with the importance of inland water bodies (rivers and lakes) in overall C balance of the terrestrial biomes, as also demonstrated in Western Siberia [102]. However, at the present status of the research, a straightforward comparison of the data on the C pattern revealed in the Sayan–Altai region and those reported in the adjacent subarctic and mountain territories requires more thorough assessment of possible controlling factors such as local climate variation, productivity of the terrestrial compartments, underground influx, respiration of sediments and primary productivity in the river and lakes. In particular, it was beyond the scope of the present study to address the soil type and chemical weathering within each individual catchment of a river or lake. Future research will focus on relating the GIS-based specific watershed parameters (including soils, vegetation, climate and lithology) and $CO_2$ emissions pattern, as it was recently developed for a number of rivers in Siberia (e.g., refs. [2,72,74]).

## 5. Conclusions

A thorough, first-time assessment of the contemporary status of the C biogeochemical cycle (concentration in and emission from the water surfaces) was conducted in lakes and rivers of the Central Asian Sayan–Altai Mountain system in order to provide a background for judging possible future changes induced by particulate climate instability in this region. Using a physicogeographical and climatic transect of inland water bodies, which comprised a large variety of natural ecosystems and landscapes of the region, we found that permafrost exerts the largest impact on lake water hydrochemistry and C pattern including $CO_2$ exchange with the atmosphere, whereas the size of the river watershed had relatively little impact on the $CO_2$ pattern. In contrast, $pCO_2$ decreased with an increase in lake size, which could be linked to a combination of factors, notably (*i*) enhanced input of terrestrial, biolabile DOM in small lakes which served as a source of $CO_2$ production in the water column; (*ii*) terrestrial input of $CO_2$-rich shallow groundwater and soil waters, more pronounced in small lakes; and (*iii*) $CO_2$ uptake in large lakes due to macrophytes, periphyton and phytoplankton activity. The oxygen regime was found to be an important controlling factor of the $pCO_2$ level in both rivers and lakes during specific seasons, likely due to sediment respiration processes. However, despite these qualitative features of possible $CO_2$ regime control in the studied water bodies, a pairwise correlation analysis did not demonstrate statistically significant relationships between $CO_2$ flux and physicogeographical parameters of river watersheds, river and lake size and internal parameters of the water

column, including basic hydrochemical characteristics, DOM concentration and quality and bacterial concentration. A likely reason is the complexity of the 'external' and 'internal' factors controlling the $CO_2$ exchange between water surfaces and the atmosphere and their frequent counteraction of $CO_2$ production/uptake in the water column, sediments and the watershed.

We argue that further assessment of possible controlling factors such as local climate variation, productivity of the terrestrial compartments, underground influx, respiration of sediments and primary productivity in the rivers and lakes are necessary for comparison of the aquatic C pattern revealed in the Sayan–Altai region in this study to those reported in the adjacent subarctic and mountain territories.

**Supplementary Materials:** The following supporting information can be downloaded at: https://www.mdpi.com/article/10.3390/w15193411/s1, Figure S1. Box plot of median and IQR range (with outlies as dots) of electrical conductivity in rivers (a); lakes (b); DIC concentrations in rivers (c); and lakes (d), averaged across seasons. Figure S2. Boxplots (median and IQR range) of electrical conductivity during different seasons (lakes and rivers combined together). Figure S3. Plots of DIC concentration (mg L$^{-1}$) as a function of electrical conductivity in large rivers (a), small rivers (b) and lakes (c). Figure S4. The relationship between $pCO_2$ and oxygen saturation degree of lake and river waters in the Tyva region (all seasons together). Figure S5. The relationship between (a) $pCO_2$ and (b) $fCO_2$ and thermokarst lakes water surface area of the Tyva region. Figure S6. The relationship between $pCO_2$ (a) and $fCO_2$ (b) in river watershed area in the Tyva region (all seasons together). Table S1. Pearson correlation coefficients between major hydrochemical and microbiological parameters of the water column, including pH, $O_2$, S.C., DOC, DIC and optical parameters of DOM reflecting DOM quality. $S_{area}$, S.C., and TBC represent area of the lake surface or river watershed, Electrical Conductivity, and Total Bacterial Count, respectively.

**Author Contributions:** Conceptualization, A.A.B., L.G.K., A.O.K. and S.K.; methodology, I.I.K., T.V.R., A.S.P., I.V.L., Z.N.K. and S.N.V.; software, I.I.K.; validation, S.N.V. and O.S.P.; formal analysis, A.A.B. and L.G.K.; investigation, A.A.B., L.G.K. and I.I.K.; resources, A.O.K. and S.K.; data curation, A.A.B. and L.G.K.; writing—original draft preparation, A.A.B. and L.G.K.; writing—review and editing, O.S.P.; visualization, A.A.B. and L.G.K.; supervision, S.K.; project administration, A.O.K.; funding acquisition, S.K. and A.O.K. All authors have read and agreed to the published version of the manuscript.

**Funding:** This research was funded by Tomsk State University Development Program "Priority 2030" (A.A.B., L.G.K., I.I.K., T.V.R., A.S.P., S.N.V., O.S.P., S.K.) and grant RSCF № 23-14-20015 (A.O.K.). Research was performed using the equipment of the unique research installation 'System of experimental bases located along the latitudinal gradient' with financial support Ministry of Science and Higher Education of Russia (RF-2296.61321X0043, 13.UNU.21.0005, contract No. 075-15-2021-672).

**Data Availability Statement:** Full data set of this study is available at the Mendeley Data Portal: Byzaakay, Arisia; Лариса, Колесниченко; Pokrovsky, Oleg (2023), "Dissolved carbon concentrations and emission fluxes in rivers and lakes of Central Asia (Sayan-Altai mountain region, Tyva)", Mendeley Data, V1, https://data.mendeley.com/datasets/d35ctw7tb3/1, accessed 23 September 2023.

**Conflicts of Interest:** The authors declare no conflict of interest.

## Appendix A

*Detailed description of rivers and lakes of the Tyva Republic sampled in this work.*

The Yenisei River is one of the longest and deepest rivers in the world and Russia, flowing into the Kara Sea of the Arctic Ocean. The length is 3487 km, the catchment area is 2,580,000 km$^2$, and the annual flow is 624.41 km$^3$. The flow velocity is from 0.3 to 5 m/s, depending on the location of the river [103]. The research was carried out in the upper reaches of the river. The river originates from the confluence of two sources—the Big Yenisei (Biy-Khem) and the Small Yenisei (Kaa-Khem). The main tributaries in the upper reaches are Elegest, Khemchik, Us, Kantegir. The city of Kyzyl and the Service-Khem kozhuun of the Republic of Tyva, together with the administrative center of Shagonar, is located in the river basin. The non-flooded channel of the Upper Yenisei (Service-Khema)

on the territory of Tyva runs through the flat terrain of the Tuva basin with a pronounced steppe landscape, has many channels and islands [104].

The basin of the Bolshoy Yenisei River (Biy-Khem) is located in the wettest, taiga, teeming with lakes and wetlands of Tyva. In the center of it is the vast Todzhinsky basin. The Biy-Khem basin, in the area from the mouth of the Seiba to the mouth of the Uyuk, is characterized by the presence of high mountains with steep slopes and weak forest cover. Below, before the confluence with the Maly Yenisei River (Kaa-Khem), smoothed open landforms and a wide river valley are typical. The length of the river is about 560 km. The width of the riverbed The Yenisei varies from 20-80 m in the upper reaches to 120–290 m in the middle and lower reaches, depths, respectively, from 1–1.5 m to 1.5–4 m. The flow velocity varies from 1.4 to 2.4 m/s. The average long-term water consumption of the Yenisei River in the closing alignment (Kara—Haak village) is 594 $m^3$/s [104].

The Maly Yenisei River (Kaa-Khem) is formed by the confluence of two rivers: Kyzyl-Khem and Balyktyg-Khem. The source of Kyzyl-Khem is located on the territory of Mongolia, whereas Balyktyg-Khem originates from the northern slopes of the Sengilen Highlands. In the upper reaches of the Yenisei River, it is a typical mountain river, its bed abounds with rocky ledges that form numerous rapids and shivers. After entering the Ulugh-Khem basin, it flows in low steppe shores. Compared to Biy-Khem, Kaa-Khem has fewer tributaries. The right tributaries of the river are Unzhey, Honga, Uzhep, Derzyg; the left ones are Shivey, Sizim, Buren. The river is quite full-flowing, fast, the speed is 1.8–2.3 m/s, the width of the channel is from 20 to 80 m. The water discharge is highly variable and depends on the amount of precipitation (Table 1). In May–early June, the spring flood comes, and in July-August, as a result of heavy rains, sometimes there is a summer flood. Groundwater takes part in feeding rivers all year round. The depth of groundwater is about 8-10 m, in floodplains of rivers and streams they come close to the surface [105].

The Tes-Khem River has a mountainous character, belongs to the Ubsu-Nur water system. The length of the river is 139 km, the width of the river is 10–100 m, the catchment area is 4390 $km^2$. The river's feeding is predominated by rain. In summer, there is a high probability of rain floods. In winter, the river freezes. The river originates on the Tannu-Ola mountain range. The mouth of the river is at an altitude of 1067 m. To the west of Tes-Khem, beyond the sands of Tsugeer-Els, there is the lake Tore-Khol, formed by the damming of the former tributary of Tes-Khem.

The Khemchik River originates on the eastern slope of the Kozer ridge from a peak of 3122 m, belonging to the Shapshalsky ridge system, on the border with the Altai Republic. The entire river is located on the territory of the Republic of Tyva. It flows between two mountain systems—the Western Sayan from the north and the Western Tannu-Ola from the south, collecting all its runoff from them. The length of the river is 320 km, the catchment area is 27 thousand $km^2$. The average water discharge is 102 $m^3$/s [104]. All sources and tributaries are fed by runoff from high ridges, and therefore the river system in the Khemchik basin is widely used for irrigation [106]. The valley is narrow, with steep banks, and there are many boulders in the riverbed. In the Khemchik basin, the river has a flat character with bends. Large stone remains are not uncommon. The flow velocity is low. The river is essentially fed from underground discharge. The Khemchik's flood occurs in summer, from June to August inclusive (with a maximum in July), and is the result of summer precipitation in the form of rain. In September, water consumption decreases significantly, and this decrease continues until the beginning of winter, along with a decrease in precipitation [107].

The Alash River, a left tributary of the Khemchik River, is formed by the confluence of the Chulcha and Kara-Khol rivers; its length is 125 km and the slope is up to 3.5 m/km. The catchment area is 4630 km. The width of the riverbed is up to 60–90 m. The valley of the river is wide. The river acquires a mountain-steppe character. Absolute elevation marks in the area of the river flow range from 1000 to 1200 m [108].

The Ak-Sug River (in the upper reaches of the Ak-Khem), a left tributary of the Khemchik River, originates in one of the lakes of the Dashtyg-Khem ridge at an altitude of 2115 m above sea level. The length of the river is 160 km, the area of its catchment area is 3170 km$^2$. The average annual water discharge is 14 m$^3$/s. Agriculture with irrigation is developed in the valley. Annual water consumption is 1% of the annual flow of the river (15 million m$^3$). The river has 46 tributaries less than 10 km, the total length of which is 173 km, there are also 82 lakes in the catchment area, the total area of which is 10.23 km$^2$. The Kyzyl-Taiga Mountain is located at the headwaters of the Ak-Sug River [109]. The absolute heights of the terrain through which the Ak-Sug River flows vary significantly from 1320 to 2100 m. The width of the riverbed ranges from 2–5 m in winter, to 15–20 m in summer, the depth, respectively, from 0.2–0.3 m to 0.7–1.1 m. The river valley on the investigated section of the Ak-Sug river is forested, the lower parts of the slopes are relatively flat, the upper ones with rocky and shrubby tundra are steep and rocky. At the beginning of the lower third of this segment on the left bank of the Ak-Suga at absolute altitudes of 1350–1500 m there is an eponymous deposit of molybdenum-copper sulfide ores. The deposit was exposed by erosion and subjected to the movement of part of the ore material by a glacier [110].

The Chadan River is a right tributary of the Khemchik River. The source is located on the western ridge of Tannu-Ola. The length of the watercourse is 98 km, the catchment area is 2200 km$^2$. The absolute height of the terrain is 800 m [111].

The Durgen River is a left tributary of the The Yenisei of the 3rd order: flows into the Mezhegey River, a tributary of the Elegest River. The source is located in the highlands of the East Tannu-Ola ridge, then the river flows along its northern macro slope. According to the State Water Register, the total length of the river is 93 km. The speed of the river flow changes in accordance with the change in terrain. The river is distinguished by a rare cascade of waterfalls. The river is located within the Mezhegeysky coal deposit (Khayan et al., 2014) [112]. In the middle course, the river flows through the territory of two villages in which there is no sewerage, the river is used for watering livestock. Part of the catchment area is occupied by a specially protected natural area—the Durgensky Nature Reserve.

The Chaa-Khol River, a right tributary of the The Yenisei River, flows through the territory of the Ulug-Khem and Chaa-Khol kozhuuns of Tyva. The total length of the river is 90 km. The catchment area has 1730 km$^2$. It begins on the northern slope of the Western Tannu-Ola ridge between the Tyndy-Ula and Hule-Bozh mountains. It flows in a northerly direction through a valley overgrown with larch forest in the upper reaches and across the steppe in the lower reaches. In the middle course it is divided into several channels. It flows into the Yenisei River at a distance of 3315 km from the mouth. The height of the mouth is 540 m above sea level [103].

The Khule River (Torgalyg), a right tributary of the Shagonar River, originates on the northern macroscline of the Eastern Tannu-Ola, flows through the Central Tuva Basin. The length of the river is 40 km, the width varies from 0.5 to 1.5 m, the catchment area is 610 km$^2$. Altitude above sea level: 535 m [113].

Anyyak-Chyrgaki River, a tributary of the Chirgaki River, a right tributary of the Khemchik River. The length of the watercourse is 52 km, the catchment area is 410 km$^2$. The bottom of the river is pebbly. During the study period, the water is transparent to the bottom, its color is bluish-green, the depth of the river ranges from 0.2 to 2.0 m. The absolute elevation of the terrain is 800 m. The source of the river begins on the north-western forested slopes of the Western Tannu-Ola [114].

The Biche-Bayan-Gol River is a right tributary of the Yenisei, into which it flows 12.5 km from the mouth, 14 km northwest of the city of Kyzyl. The length of the watercourse is 32 km [115].

The Adyr-Khem River flows through the territory of the Barun-Khemchik district on the Alash plateau. The width of the Adyr-Khem River is 1.5 m, the depth is from 0.25–1 m, the flow velocity is about 0.25 m/s [116].

Lake Tore-Khol is the only large freshwater lake in the Ubsunur basin. The lake has no tributaries, receiving water from springs of sand dunes, which are located in a horseshoe-shaped bowl of the southern shore and drain into the lake in the form of a stream. The Torehole is located in a shallow depression with flat sides, the average depth is 7 m, the area is 42 km$^2$. The greatest depth (up to 40 m) was noted at the isthmus in the southern part. The absolute height of the terrain is 1149 m. The lake is oligotrophic, the least productive of all the lakes of the Ubsunur basin [117]. It contains a relatively small percentage of organogens and bacteria. Therefore, the lake is transparent, clean, rich in underutilized oxygen by organisms. In the summer, the surface layers warm up well, in the middle of August in the afternoon the temperature rises to 21 °C at an air temperature of 32 °C [118].

Lake Chagytai is the largest freshwater lake of the Tuva basin, located in its southern part on the border of the settled hilly foothills of the Vostochny Tannu-Ola ridge, at an altitude of 1003 meters above sea level. This is the largest freshwater lake in the Tuva basin. It has the shape of an almost regular circle with a diameter of about 6 km, the length of the coastline is about 20 km. The area of the lake is 2860 hectares, its maximum depth reaches 17 meters [103]. In the west and southeast of the reservoir, the slopes of the mountains are covered with taiga. The bottom of the lake is sandy and pebbly. The shores are mostly flat, sometimes rocky, sometimes sandy. The south-eastern shore is swampy, overgrown with talus, birches, larches. The only river flowing from the lake, the Mazhalyk River, originates here.

Lake Cheder is located 45 km south of the city of Kyzyl, in the south of the Tuva basin in a drainless depression, on board of which the sandy-clay rocks of the Jurassic come out. The depression is surrounded by a hilly, treeless plain. The lake is drainless, located in a shallow depression, within the vast Central Tuva basin, at an absolute mark of 706 m. The lake is fed by a stream flowing into the lake in the southern part, as well as groundwater from quaternary lake sediments. The lake has a slightly elongated shape. The water surface area is 430 ha, length—4.5 km, width—0.8–1.5 km, depth—1.5–1.8 m [119]. The water in the lake is highly mineralized. The main component is sodium sulfate. The mineralization of water in the lake, depending on the level, ranges from 80 in April and May to 200 grams per liter in August. Potassium salts (0.250 g/L), fluorine (0.003 g/L), iodine (0.001 g/L), strontium (0.001 g/L) are present in small quantities. The shores and bottom of the lake are composed of silt mud with a thickness of up to 2 meters. The mud is gray or gray-black with the smell of hydrogen sulfide. On the territory of the lake there is 46-m deep well with mineral drinking water. The mineral water captured by the well is cold, low-mineralized chloride magnesium-sodium-calcium in composition and slightly alkaline in the nature of the reaction of the medium [120].

Thermokarst lakes are located in the Alash Highlands in the Barun-Khemchik district, 6.5 km from the border with Khakassia and 150 km from the city of Ak-Dovurak. Absolute marks in the area of lakes range from 1800 to 2000 m. Absolute heights in the lakes range from 1840 to 1860 m. There are about 200 thermokarst lakes in the study area. Basically, the lakes are small, most of the lakes have a rounded, oval-elongated shape, some of the shores are overgrown and swampy. Two round-shaped lakes were selected for the study.

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
