# Peer review of "Dissolved Carbon Concentrations and Emission Fluxes in Rivers and Lakes of Central Asia (Sayan–Altai Mountain Region, Tyva)"

_water, doi:10.3390/w15193411_

Round 1

Reviewer 1 Report

This carried out systematic measurements of dissolved C and CO2 emissions in 15 rivers and 5 lakes located along a macro-transect of various natural landscapes in the Sayan-Altai mountainous region. It also tested  the impact of permafrost, river watershed size, lake area and climate parameters as well as internal biogeochemical drivers on CO2 concentration and emissions in lakes and rivers of this region. This study conducted abundant field investigations to obtain field data to support the study results. I’ve check the results, most of which are reliable. It is suitable for the publication in the journal of WATER. However, it also can be improved to be more flawless. Here are my suggestions:

1. The paper’s title is too long. It is recommended to use a relatively concise title.

2. The abstract of the paper simply said what the study has done. It should be more important to explain what important findings were obtained in this study.

3. The authors said that many water chemistry indicators were collected during the field survey of the study, such as pH value, conductivity, bacterial population, etc. Did they also collect wind speed? Wind speed has a significant impact on CO2 emission fluxes. If wind speed considered, more statistically significant relationship could be obtained in Table 5. Also, was flow velocity collected during investigation?

4. Soil type, and associated different weathering processes can be taken into account when discussing drivers of CO2 fluxes.

5. I know this is an initial assessment across permafrost, but this data can be drilled down further to obtain more meaningful findings.

N/A.

Reviewer 2 Report

The manuscript represents a very urgent and modern study, focused at approaches for the estimation of a number of environmental factors on distribution of the key biogeochemical parameters of a carbon cycle in very interesting geographic setting  (Sayan-Altay Mountain Region). The CO2 content and emission was chosen as an integral parameter, assuming its obvious significance for climate change.The natural variability of the key biogeochemical parameters have been assesed and interpreted across the vast area, including the interconnected water bodies. 

The comments and suggestions are specified in the file attached as the pdf cooments
